# Stability Assessment of Four Chimeric Proteins for Human Chagas Disease Immunodiagnosis

**DOI:** 10.3390/bios11080289

**Published:** 2021-08-22

**Authors:** Paola Alejandra Fiorani Celedon, Leonardo Maia Leony, Ueriton Dias Oliveira, Natália Erdens Maron Freitas, Ângelo Antônio Oliveira Silva, Ramona Tavares Daltro, Emily Ferreira Santos, Marco Aurélio Krieger, Nilson Ivo Tonin Zanchin, Fred Luciano Neves Santos

**Affiliations:** 1Molecular Biology Institute of Paraná, Curitiba, Paraná 81350-010, Brazil; paolafc@ibmp.org.br (P.A.F.C.); udoliveira@ibmp.org.br (U.D.O.); marco.krieger@fiocruz.br (M.A.K.); 2Gonçalo Moniz Institute, Oswaldo Cruz Foundation, Salvador, Bahia 40296-710, Brazil; leonardo.leony@fiocruz.br (L.M.L.); natalia.erdens@fiocruz.br (N.E.M.F.); angelo.oliveira@fiocruz.br (Â.A.O.S.); ramona.daltro@fiocruz.br (R.T.D.); emily.santos@fiocruz.br (E.F.S.); 3Carlos Chagas Institute, Oswaldo Cruz Foundation, Curitiba, Paraná 81350-010, Brazil; nilson.zanchin@fiocruz.br; 4Integrated Translational Program in Chagas Disease from Fiocruz (Fio-Chagas), Vice Presidency of Research and Biological Collections, Oswaldo Cruz Foundation, Rio de Janeiro, Rio de Janeiro 21040-900, Brazil

**Keywords:** Chagas disease, immunoassays, chimeric proteins, stability

## Abstract

The performance of an immunoassay relies on antigen-antibody interaction; hence, antigen chemical stability and structural integrity are paramount for an efficient assay. We conducted a functional, thermostability and long-term stability analysis of different chimeric antigens (IBMP), in order to assess effects of adverse conditions on four antigens employed in ELISA to diagnose Chagas disease. ELISA-based immunoassays have served as a model for biosensors development, as both assess molecular interactions. To evaluate thermostability, samples were heated and cooled to verify heat-induced denaturation reversibility. In relation to storage stability, the antigens were analyzed at 25 °C at different moments. Long-term stability tests were performed using eight sets of microplates sensitized. Antigens were structurally analyzed through circular dichroism (CD), dynamic light scattering, SDS-PAGE, and functionally evaluated by ELISA. Data suggest that IBMP antigens are stable, over adverse conditions and for over a year. Daily analysis revealed minor changes in the molecular structure. Functionally, IBMP-8.2 and IBMP-8.3 antigens showed reactivity towards anti-*T. cruzi* antibodies, even after 72 h at 25 °C. Long-term stability tests showed that all antigens were comparable to the control group and all antigens demonstrated stability for one year. Data suggest that the antigens maintained their function and structural characteristics even in adverse conditions, making them a sturdy and reliable candidate to be employed in future in vitro diagnostic tests applicable to different models of POC devices, such as modern biosensors in development.

## 1. Introduction

Chagas disease is a deadly, neglected, tropical infection caused by the hemoflagellate parasite *Trypanosoma cruzi*. According to the World Health Organization, 5.7 million individuals are infected by the parasite, resulting in 7500 deaths annually, mostly in the continental Western Hemisphere, resulting in a massive disease burden in the 22 endemic countries [1]. The parasite can be transmitted through several pathways, such as by contact with excrements from infected triatomine bugs (hematophagous insects of the Triatominae family), consumption of contaminated beverages and food, from mother-to-child during pregnancy, whole blood or blood derivatives transfusion, tissue and organ transplantation and through laboratory work accidents [2].

Two distinct phases occur during the natural course of Chagas disease progression. The initial acute phase is characterized as an unspecific oligosymptomatic febrile illness. Infected individuals present high parasitemia, which enables the parasitological diagnosis, based on the direct visualization of the parasite in a thick blood smear. The acute phase lasts for 2–3 months after initial infection. This is followed by the gradual resolution of the clinical manifestations (when present) and the start of the lifelong chronic phase, characterized by an intermittent or absent parasitemia, as well as high levels of IgG anti-*T. cruzi* antibodies [2]. As such, direct methods are unacceptably accurate as diagnostic methods in the chronic stage of the disease, thus, in vitro diagnostics (IVD) based on indirect immunoassays are overwhelmingly recommended. Indirect IVD methods are based on the detection of specific antibodies produced against a certain pathogen, such as indirect immunofluorescence (IIF), indirect hemagglutination (IHA), rapid diagnostic tests (RDT) and enzyme-linked immunosorbent assay (ELISA). Such serological methods rely on the interaction between the antibodies from the patient and epitopes from the pathogen antigen, as such, the structural integrity of those antigens is paramount for efficient antibody binding. Furthermore, the stability of the antigens over time and different environmental conditions must be taken into consideration. In a laboratory setting, the shelf and storage lifespan can be determinant when choosing the appropriate reagent for the routine and when point-of-care testing is being considered, an antigen must retain its structural integrity over a prolonged time, despite significant temperature variations, as can be expected between storage and transportation.

Among the available commercial IVD tests to identify chronic Chagas disease, ELISA and RDT are the most used due to their low cost and overall efficiency. However, these methodologies regularly present inconsistent performance, which is attributed to various distinct reasons, such as the chosen capture antigens [3], the varying degree of immune responses against the infection [4,5], *T. cruzi* high genetic and phenotypic intraspecific diversity [6] and variation in disease prevalence [7,8]. Accordingly, the World Health Organization (WHO) recommends the use of two different serological tests in parallel to diagnose Chagas disease in humans [9]. The use of chimeric proteins, composed of conserved immunodominant, and are tandemly repeated sequences of several different antigens of *T. cruzi*, can be a strategy to address the inconsistent performances of IVD tests [10,11,12]. This strategy can also address issues commonly attributed to recombinant proteins, such as the lower sensitivity in comparison to lysates or native antigen mixtures, which is attributed to a lower epitope diversity. The use of chimeric proteins in IVD simultaneously addresses the lack of reproducible performance parameters, while increasing both sensitivity and specificity, as a result of the greater diversity of distinct and conserved immunodominant epitopes, from numerous antigens, which are presented in these proteins. Chimeric proteins have also been employed in IVD [13,14,15] or as a vaccine [16,17] for other infectious diseases. Considering the predicaments herein set forth, our group expressed four *T. cruzi* chimeric proteins, called IBMP-8.1, IBMP-8.2, IBMP-8.3 and IBMP-8.4 (Molecular Biology Institute of Paraná—IBMP in Portuguese acronym), and assessed their performance in diagnosis chronic Chagas disease in dogs [18,19] and humans, from several endemic and non-endemic settings from Brazil [20], Argentina, Paraguay, Bolivia [21] and Spain [22].

The ability of immunoassays to detect specific antibodies depends on the spatial distribution and availability of epitopes on the solid phase. Although peptides sequences with conformational preferences have been shown to be preferentially recognized by antibodies against native protein epitopes, some evidence supports the idea that IBMP antigens are composed of linear epitopes [10]. Aggregate formation, degradation, or even conformation changes can lead to the impairment of antigen-antibody detection, caused by hidden or folded epitopes, hinder antigen accessibility, thereby leading to misdiagnosis. However, it is not clear whether these conditions, which are often the result of improper storage and handling, could impact the assay’s performance when chimeric proteins composed of linear epitopes are employed as solid-phase antigens.

Likewise, due to IVD tests tendency to evolve towards automated and miniaturized microfluidic systems, it is pertinent to obtain data regarding the feasibility of novel molecules towards integrated IVD devices. Protein structural conformation, protein-protein interaction and stability assessment are paramount for an efficient immunoassay, whether it is applied in an ELISA system or microfluidic devices. Recently, the IBMP molecules performance profile for chronic Chagas disease diagnosis was similar to that of an impedimetric immunosensor using dual screen-printed carbon electrode [19], although these interactions in turn occurs mostly in liquid phase-like mode.

In this work, in order to evaluate the outcome of linking several epitopes separated by artificially designed spacers, we conducted a functional and structural analysis of four chimeric antigens for chronic Chagas disease in vitro diagnosis. We also assessed the impact of different antigen buffers and adverse environmental conditions on the performance of the immunoassays. Bearing in mind that the formation of secondary structures, dimerization and agglutination negatively impacts the performance of an indirect immunoassay from exploring linear epitopes, and that antigens are the main gear driving indirect serological methods, the analysis regarding an antigen’s behavior at different environmental conditions, temperatures and buffers paves the way towards a robust antigen preparation for a reliable and biosensor.

## 2. Materials and Methods

### 2.1. Study Design

The study was carried out using both dissolved and adsorbed antigens (Figure 1). To stringently evaluate the extent of structural perturbations in different environmental conditions, thermic and long-term stability at 4 °C was analyzed. Initially, samples were heated up to 85 °C and cooled to 4 °C to verify the reversibility or irreversibility of heat-induced denaturation (time exposure in each temperature: 1 min, 10 min and 20 min). Therefore, samples were taken before, and after, the tests and analyzed by circular dichroism (CD), dynamic light scattering (DLS), sodium dodecyl sulfate-polyacrylamide gel electrophoresis (SDS-PAGE) and enzyme-linked immunosorbent assay (ELISA). With respect to storage stability of soluble antigens at room temperature (25 °C), we analyzed the samples at time zero (control) and aliquots were drawn out every 24 h for 72 h. Antigen structure was analyzed by CD, DLS, SDS-PAGE, and functionally assed through ELISA. Long-term stability tests at 4 °C were performed over 1 year (Figure 1). Accordingly, eight sets of microplates were sensitized with IBMP antigens and stored at 4 °C in hermetically sealed storage bags with desiccant. Similarly, eight sets of *T. cruzi*-positive and negative sera were aliquoted and stored at −20 °C until analysis functional and structural assessment through ELISA.

### 2.2. IBMP Protein Expression and Purification

Antigen sequence selection (Table 1), gene construction and recombinant expression were described in Santos et al. [10]. Briefly, *T. cruzi* synthetic genes were subcloned into the pET28a vector (Novagen, Madison, WI, USA) and expressed as soluble proteins in *Escherichia coli* BL21-Star (DE3) cells grown in LB medium supplemented with 0.5 µM isopropyl-β-D-1-thiogalactopyranoside (IPTG). The antigens were purified by affinity and ion-exchange chromatography using columns and chromatographers supplied by GE Healthcare. Finally, purified proteins were quantified using a fluorometric assay (Qubit 2.0, Invitrogen Technologies, Carlsbad, CA, USA).

### 2.3. Gel Electrophoresis

Samples (1 µg) were resuspended in loading buffer, subjected to electrophoresis in SDS–PAGE [23] (8 × 9 cm and 0.75 mm thick, 12% polyacrylamide) and stained with Coomassie brilliant blue-250 for visualization. SDS-PAGE was used to evaluate the protein degradation of soluble antigens during analysis.

### 2.4. Circular Dichroism (CD) Measurements

To further evaluate thermal impact over structural content and conformation of IBMP antigens, far-UV CD spectra were assessed on a CD spectrophotometer, (Jasco J-815, Tokyo, Japan), equipped with a Peltier type CD/FL cell thermoelectric sample holder (Jasco, Tokyo, Japan). Samples were analyzed at 0.2 mg/mL in a 1-mm path length quartz cuvette. Each spectroscopic readout represented the average of four continuously acquired scans. All readouts were rectified considering the buffer background noise. Each scan within 193–260 or 202–260 nm range was acquired through a scanning rate of 100 nm/min, 0.5 nm data pitch, 1 nm bandwidth, and 1 s of data integration time. The results were expressed in molar ellipticity [θ]_λ_ (deg × cm^2^ × dmol^−1^) and shifts at 208–222 nm and 215 nm were used to assess α-, and β-structural content, respectively. Protein secondary structure analysis was performed using the Dichroweb platform.

### 2.5. Dynamic Light Scattering (DLS)

In order to determine the hydrodynamic radius (Rh) and polydispersity, DLS measurements were undertaken utilizing a DynaPro NanoStar Dynamic Light Scattering analyzer (Wyatt Technology Corp., Santa Barbara, CA, USA), equipped with a Ga-As laser (120 mW), operating at a wavelength of 658 nm. Protein samples were placed in a 1-mm^2^ quartz cuvette and measurements were performed at RT (22–25 °C). Data analysis was carried out using the software Dynamics v.7.1.7.16.

### 2.6. Clinical Specimens and Indirect ELISA

Anonymized human sera samples from individuals infected *(n* = 46) or non-infected (*n* = 46) with *T. cruzi* were obtained from the Chagas Disease Reference Laboratory (Aggeu Magalhães Institute; Oswaldo Cruz Foundation, Pernambuco, Brazil) and used to assess the reactivity of IBMP antigens associated with the three different conditions (Figure 1). Sample characterization was based on the concordance between two distinct serological tests for Chagas disease, as recommended by the World Health Organization [9]. Assays were performed according to the procedure described previously [10]. Briefly, IBMP antigens were diluted at 12.5 ng (IBMP-8.2) and 25.0 ng (IBMP-8.1, IBMP-8.3 and IBMP-8.4) in carbonate buffer (0.05 M, pH 9.6), then 96-well microplates (Nunc, Roskilde, Denmark) were coated with 100 µL per well and subsequently blocked with Well Champion reagent (Kem-En-Tec, Taastrup, Denmark) according to the manufacturer’s instructions. Sera samples were pre-diluted at 1:100 in 0.05 M phosphate-buffered saline (PBS; pH 7.2) and 100 µL was transferred to each well. After incubation at 37 °C for 60 min, the microplates were washed with PBS-0.05% Tween 20 (PBS-T). HRP-conjugated goat anti-human IgG (Bio-Manguinhos, FIOCRUZ, Rio de Janeiro, Brazil) was diluted in PBS at 1:40,000, and then 100 µL was transferred to each well and incubated for 30 min at 37 °C. Following the incubation and washing, 100 μL of TMB substrate solution (tetramethyl-benzidine; Kem-En-Tec, Taastrup, Denmark) was added and incubated at room temperature for 10 min in the dark. Finally, the reaction was stopped with 50 μL of 3 N H_2_SO_4_, and the optical density was measured at 450 nm in a Multiskan^®^ FC microplate spectrophotometer (Thermo Scientific^TM^, Ratastie, Finland).

### 2.7. Data Analysis

Geometric mean ± SD calculation was measured for all variables. In order to verify data normality, the Shapiro-Wilk test was performed, followed by Student’s T test. Whenever data variance homogeneity assumption couldn’t be confirmed, the Wilcoxon signed-ranks test was employed. All statistical significance analyses had a two-tailed distribution and a *p*-value under 5% was considered significant (*p* < 0.05). Regression curve analysis was performed for each temperature condition using positive/negative (P/N) ratio values and protein exposure period for each thermal reading point. ELISA reactivity data at different thermal conditions were subjected to statistical analysis employing student’s t-test to assess significance level. Cut-off value analysis was defined by the largest distance from the diagonal line of the receiver operating characteristic curve (ROC) (sensitivity × (1-specificity)) to identify the optimal ELISA OD value that best differentiates between negative and positive samples. The confidence interval (CI) was developed to address the proportion estimates precision with a confidence level of 95%. Data were examined using GraphPad Prism version 8 (San Diego, CA, USA).

## 3. Results

### 3.1. Reversibility of Heat-Induced Denaturation

Since changes in CD spectra and light scattering may reflect structural shifts of the molecules, we previously characterized the IBMP chimeric antigens by these methodologies to determine their typical conformation under different buffer compositions [10]. On that occasion, 50 mM carbonate-bicarbonate buffer pH 9.6 was shown to be the most appropriate buffer system for reducing protein aggregation, while maintaining the original conformation. Now, we demonstrate that when soluble in carbonate buffer, the IBMP antigens are very stable, and even after some degradation under temperature effect, little of their diagnostic capacity is lost. By monitoring CD spectra of the 4 chimeric proteins, it could be seen that denaturation-renaturation by rapid thermal stress (from 4 °C to 85 °C to 4 °C) did not cause changes relative to the original conformation for antigens IBMP-8.1, IBMP-8.2 and IBMP-8.4 (Figure 2A,E,M). Whereas, IBMP-8.3 seems to keep an unfolded conformation after heat-denaturation (Figure 2I).

In the case of IBMP-8.2, the CD signals of pre- and post-heating are very similar, but the DLS analysis shows that it has aggregated as observed by the increase in the hydrodynamic radius. There is almost no degradation, which can be visualized by the analysis by electrophoresis on polyacrylamide gels (Figure 2C,G,K,O). Despite the denaturing conditions, all antigens continue to be recognized by anti-*T. cruzi* antibodies (Figure 2D,H,L,P), suggesting that individual epitopes remained competent for antibody interaction. Individual data points of reversibility of heat-induced denaturation evaluation are available in the Appendix A.

### 3.2. Storage Stability at 25 °C

Daily analysis of the short-term assay revealed CD spectra highly similar to the original for the four proteins, indicating no changes in the conformation. The sudden increase of hydrodynamic radius (Hr) for IBMP-8.1 could represent an aggregation of the antigen while the Hr of the others remains similar to the original (Figure 3B,F,J,N). With respect to polydispersity, IBMP-8.1 seems to form homogenous aggregates and the reduced polydispersity of 8–4 at the 72 h time point may be due to degradation (Figure 3B). Gels show that IBMP-8.1 (Figure 3C) and IBMP-8.2 (Figure 3G) can hold most integrity until 72 h upon exposure to 25 °C in solution. In turn, IBMP-8.3 (Figure 3K) and IBMP-8.4 (Figure 3O) suffered severe degradation after 48 h. Although protein degradation was seen on the gels (not determined sizes), CD spectra did not change significantly at the 72 h time point for IBMP-8.3 and IBMP-8.4. The CD signal of these samples may be provided by the protein fragments still present in the samples. Functionally, IBMP-8.2 (Figure 3H) and IBMP-8.3 (Figure 3L) antigens showed reactivity to anti-*T. cruzi* antibodies, even 72 h after the beginning of the analyzes. Contrarily, IBMP-8.1 (Figure 3D) and IBMP-8.4 (Figure 3P) antigens have gradually lost their reactivity in the ELISA assays after 72 h. Individual data points of storage stability at 25 °C evaluation are available in the Appendix A.

### 3.3. Long-Term Stability Analysis at 4 °C

Long-term stability tests at 4 °C were performed using eight points of ELISA over 1 year (Figure 4). The P/N ratio of all IBMP chimeric proteins was comparable to that found for the control group (time 0), therefore, the ratio did not modify after 364 days. Exceptions for IBMP-8.1 and IBMP-8.3 antigens after 312, and 260 days, respectively. In these cases, P/N ratio slightly declined, but remained above the cut-off (Figure 4A; Individual data points are available in the Appendix A). Overall, all absorbed proteins have shown stability over 1 year. Similar results were observed when accuracy diagnostic was assessed. Considering the overlap of 95% CI values, no significant difference in accuracy was observed among all eight analysis points for the four antigens (Figure 4B).

## 4. Discussion

The high accuracy of IBMP (-8.1, -8.2, -8.3 and -8.4) chimeric antigens in diagnosing chronic Chagas disease, both in endemic and non-endemic settings has been established in previous studies [10,19,20,21,22,24,25,26,27]. However, the stability of these molecules under different stress situations was not yet known. Here, we explored the structural stability of these four antigens in different conditions using CD and DLS to gain information on the level of alterations that might take place on their structure and state of oligomerization and aggregation, respectively. Moreover, protein degradation and functional analysis have been carried out by SDS-PAGE, and indirect ELISA, respectively. Although their performance could be determined by any immunological technique, using microtiter plates, beads, nanosensors and others [28,29]. We also evaluated the stability of absorbed antigens in microplates for one year. The results obtained allow us to infer that all chimeric antigens are highly stable, preserving their functionality in immunoassays, even after exposure to extreme conditions.

Previously, our group characterized the secondary structure content and solubility of these IBMP chimeric antigen in different buffering agents: 50 mM carbonate-bicarbonate, pH 9.6; 50 mM sodium phosphate, pH 7.5; and 50 mM MES ([2-(n-morpholino) ethanesulfonic acid], pH 5.5) [10]. Those studies have demonstrated that IBMP-8.1 and IBMP-8.3 proteins predominantly comprised random coil as verified by neutral CD values between 215 and 240 nm and negative values at about 200 nm. On the other hand, CD spectra of IBMP-8.2 and IBMP-8.4 proteins exhibited a negative minimum at ~203 nm and shoulder at ~220 nm. The intensity of the CD signal at ~220 nm indicates that, in addition to random coil, IBMP-8.2 and IBMP-8.4 present also a certain content of α-helices. No conformational changes were observed in IBMP-8.1 and IBMP-8.3 coil proteins upon solubilization in different buffering agents. However, CD spectra of IBMP-8.3 and IBMP-8.4 proteins showed a slight shift to longer wavelengths upon solubilization in acidic pH, indicating changes in secondary structure content. With respect to DLS data, all proteins presented less polydisperse values (<20%) in 50 mM carbonate-bicarbonate (pH 9.6) when compared to other buffers tested. These data indicate that all proteins can be influenced by the environment. Therefore, 50 mM carbonate-bicarbonate (pH 9.6) buffer system was chosen to carry out the present study.

The chosen buffer alone conserved epitopes even during long lasting storage. The use of stabilizing compounds, which increases costs and offers interference, especially with downstream applications, is a common issue regarding antigen and antibodies storage. The reduced intrinsic structure of our molecules makes them ideal for various methodologies, where negative interferences are caused by detergents and reducing or chaotropic agents.

CD spectra on the reversibility of heat-induced denaturation assessment showed that only IBMP-8.3 chimeric protein underwent a strong change in CD signal, assuming an unfolded conformation after heating. For other proteins (IBMP-8.1, IBMP-8.2 and IBMP-8.4), CD signal suggests that these molecules have a post-heating renaturation capacity. Despite denatured, IBMP-8.3 protein did not exhibit aggregate formation as observed by DLS analysis. No significant aggregation was also observed for the IBMP-8.1 and IBMP-8.4 chimeric proteins, although there is an increase in polydispersity for IBMP-8.4. Only IBMP-8.2 presented aggregate formation. Indeed, the hydrodynamic radius substantially changed from 3.7 to 12.5 (Peak 1; %M: 4.7; Pd 12.3), 96.8 (Peak 2; %M: 5.2; Pd 12.3) and 376.3 (Peak 3; %M: 90.17; Pd 18.9). These findings indicate the formation of large protein aggregates, despite monodispersing (Pd < 20). Considering the polyacrylamide gels, no significant degradation was observed with the IBMP proteins. Despite the denaturing conditions, all antigens continue to be recognized by anti-*T. cruzi* antibodies, indicating little of their diagnostic capacity was lost. Indeed, individual epitopes remained available for antigen-antibody interaction.

At 25 °C, all IBMP chimeric proteins proved to be stable for three days (72 h), according to the absence of changes in the CD-spectra. DLS analysis also reveals that, except for IBMP 8-1, there is little change in the aggregation state of the protein samples, indicating high solubility for all proteins. SDS-PAGE, used to assess the integrity of proteins, revealed that IBMP-8.1 and IBMP-8.2 can hold some integrity until 72 h exposure at 25 °C. On the other hand, IBMP-8.3 and IBMP-8.4 suffered severe degradation after 48 h. Despite degradation, CD spectra did not significantly change for IBMP-8.3 and IBMP-8.4 at the 72 h period, suggesting that the CD signal came from short sequences with that of the corresponding full-length sequences. Nevertheless, these proteins showed high resistance to proteolysis. Daily analysis of the short-term assay by ELISA demonstrated that IBMP-8.2 and IBMP-8.3 proteins remain reactive to anti-*T. cruzi* antibodies, even 72 h after the beginning of the analysis. On the other hand, IBMP-8.1 and IBMP-8.4 lost their functionalities to recognize anti-*T. cruzi* antibodies. This is probably due to the degradation of essential epitopes used by these antibodies to bind to IBMP antigens.

Long-term stability analysis at 4 °C revealed that all proteins are stable over one year. We observed that the accuracy values did not change significantly among all eight analysis points for the four antigens. Overall, all absorbed proteins have shown stability over 364 days.

Demonstrating the optimal conditions favoring Ag-Ab kinetics can bring insights into the development of diagnostic tests, regardless of the employed platform. For example, as herein demonstrated, proteins undergo structural changes at different environments, depending on its internal interactions’ strength and the characteristics of the protein surface within the solvent/buffer. These changes certainly influence the definition of the parameters that are normally used in biosensors, such as interaction buffer, immobilization time on the electrode surface, dilution ratio, etc., as well as assist in the decision identifying the appropriate buffer for sensitizing a microtiter plate for ELISA, or activating magnetic bead’s surface in a microarray assay.

## 5. Conclusions

In this work, we are demonstrating that when diluted in carbonate buffer the IBMP-8.1 to IBMP-8.4 antigens are very stable, and even after some degradation under temperature effect. Furthermore, absorbed proteins were stable for over one year. The main limitation of this study is the lack of data in the scientific literature to compare our findings. However, we believe that the publication of stability analyses provides relevant information for the utilization of recombinant antigens in immunoassays.

## Figures and Tables

**Figure 1 biosensors-11-00289-f001:**
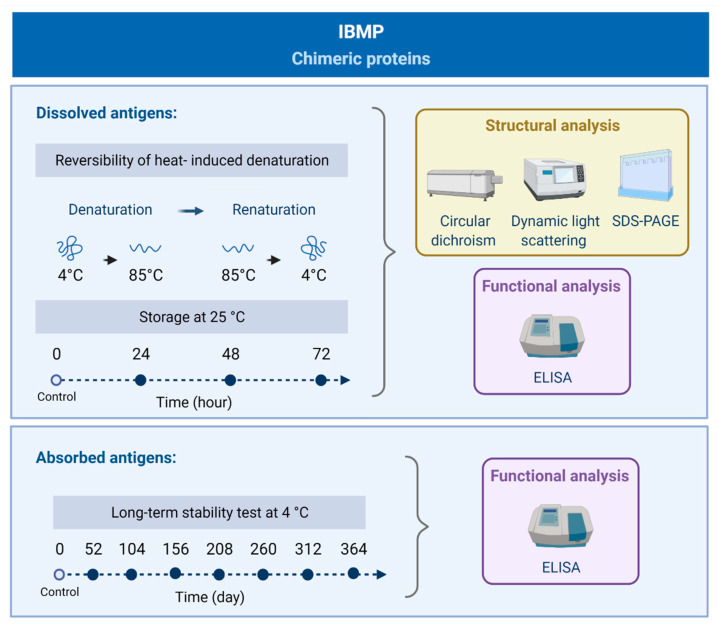
Flowchart of the study design for evaluating the stability of dissolved and absorbed IBMP chimeric proteins by using functional (ELISA) and structural (CD, DLS, SDS-PAGE) analysis.

**Figure 2 biosensors-11-00289-f002:**
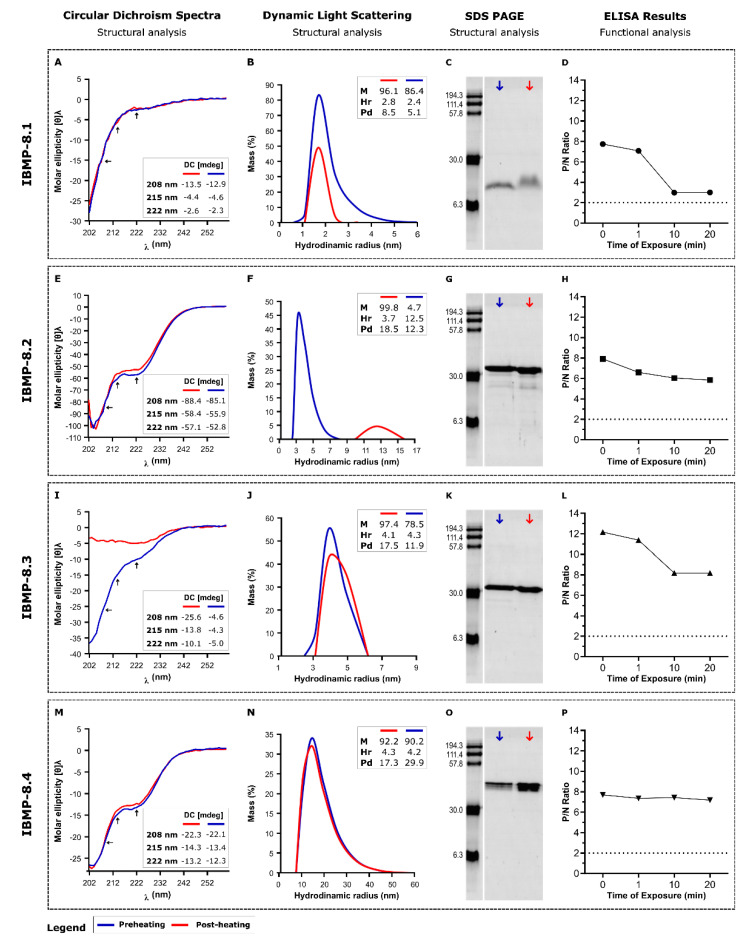
Evaluation of heat-induced denaturation reversibility for the IBMP recombinant chimeric proteins by CD, DLS, SDS-PAGE and sera reactivity by ELISA. CD (Circular dichroism); Hr (Hydrodynamic radius); M (%mass); min (Minute); nm (Nanometer); Pd (Polydispersity); P/N ratio (Positive/negative ratio). Boxes in (**A**,**E**,**I**,**M**) indicate the values of CD signal intensity at 208, 215 and 222 nm. Boxes in (**B**,**F**,**J**,**N**) indicate the values of %mass, hydrodynamic radius and polydispersity. (**C**,**G**,**K**,**O**) are the SDS-PAGE images of the antigens after the treatments. (**D**,**H**,**L**,**P**) shows the reactivity of the antigens in ELISA assay after the heat treatment.

**Figure 3 biosensors-11-00289-f003:**
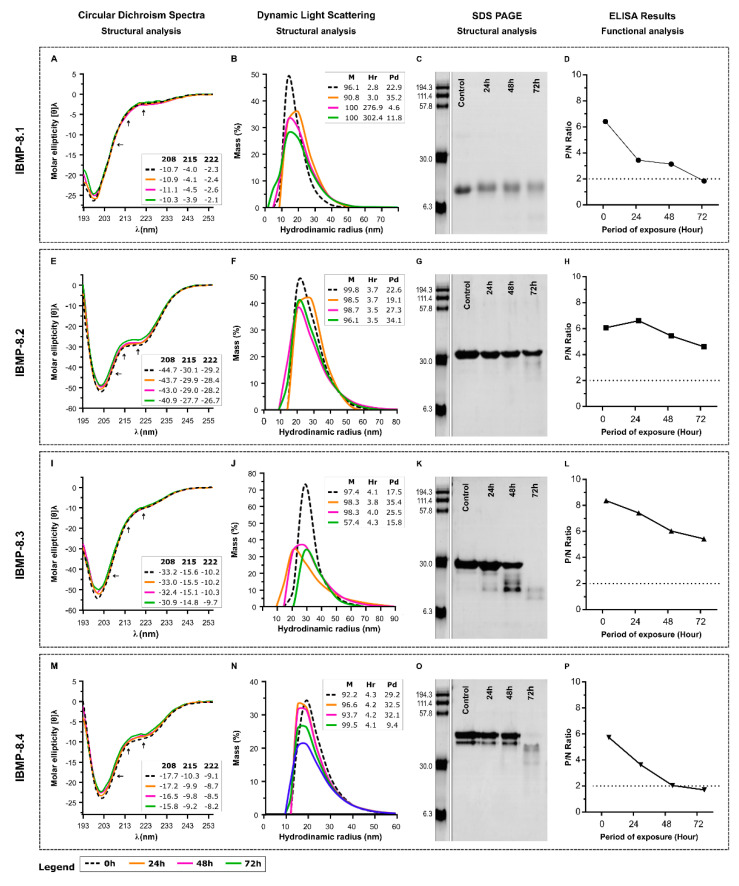
Evaluation of storage stability at 25 °C for IBMP recombinant chimeric proteins by CD, DLS, SDS-PAGE and sera reactivity by ELISA. DC (Circular dichroism); h (hours); Hr (hydrodynamic radius); M (%mass); nm (Nanometer); P/N ratio (Positive/negative ratio). Boxes in (**A**,**E**,**I**,**M**) indicate the values of CD signal intensity at 208, 215 and 222 nm. Boxes in (**B**,**F**,**J**,**N**) indicate the values of %mass, hydrodynamic radius and polydispersity. (**C**,**G**,**K**,**O**) are the SDS-PAGE images in the times described. (**D**,**H**,**L**,**P**) represent the reactivity of the antigens in ELISA assay after exposure time.

**Figure 4 biosensors-11-00289-f004:**
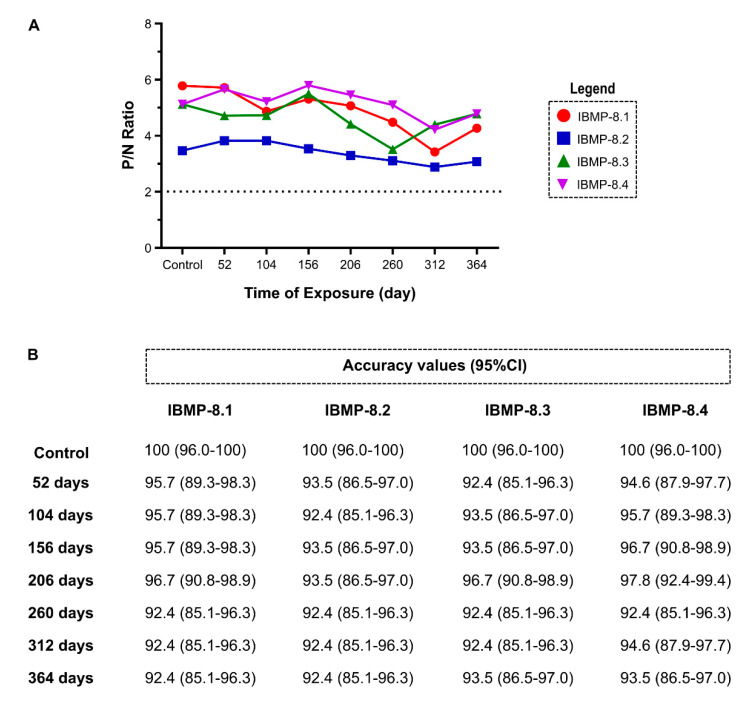
Stability of IBMP chimeric proteins over 1 year. (**A**) Ratio of positive to negative (P/N) serum reactivity in indirect ELISA; (**B**) Accuracy values for each point of analysis.

**Table 1 biosensors-11-00289-t001:** Constitution of the IBMP chimeric recombinant proteins.

Chimeric Antigen	Sequence Name	Amino Acid Range	Gene Bank Sequence ID
IBMP-8.1	Trans-sialidase	747–774	XP_820062.1
	60S ribosomal protein L19	218–238	XP_820995.1
	Trans-sialidase	1435–1449	XP_813586.1
	Surface antigen 2 (CA-2)	276–297	XP_813516.1
IBMP-8.2	Antigen, partial	13–73	ACM47959.1
	Surface antigen 2 (CA-2)	166–220	XP_818927.1
	Calpain cysteine peptidase	31–97	XP_804989.1
IBMP-8.3	Trans-sialidase	710–754	XP_813237.1
	Flagellar repetitive antigen protein	15–56	AAA30177.1
	60S ribosomal protein L19	236–284	XP_808122.1
	Surface antigen 2 (CA-2)	279–315	XP_813516.1
IBMP-8.4	Shed-acute-phase-antigen	681–704	CAA40511.1
	Kinetoplastid membrane protein KMP-11	76–92	XP_810488.1
	Trans-sialidase	1436–1449	XP_813586.1
	Flagellar repetitive antigen protein	20–47	AAA30177.1
	Trans-sialidase	740–759	XP_820062.1
	Surface antigen 2 (CA-2)	276–298	XP_813516.1
	Flagellar repetitive antigen protein	1–68	AAA30197.1
	60S ribosomal protein L19	218–238	XP_820995.1
	Microtubule-associated protein	421–458	XP_809567.1

## Data Availability

Data is contained within the article or supplementary material.

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
