# Peer review of "Stability Assessment of Four Chimeric Proteins for Human Chagas Disease Immunodiagnosis"

_biosensors, 2021, doi:10.3390/bios11080289_

Round 1

Reviewer 1 Report

This paper is dealing with the preparation and stability characterization of synthetic antigens to diagnose anti-chagas disease antibodies by ELISA. Although its importance is understandable, this topic is not related to the development of biosensor, at least in this stage. In general, immunoassays that require separation before measurement to obtain signal is not considered biosensor. Hence, here I just note some technical points to be considered in more suitable journal.

Line 90 and 95, The authors state that the epitopes of IBMP antigen are linearized or linear, while its meaning is not clear. Does it mean that these proteins are recognized even after denaturation? If it is, why the authors use CD spectra to track its secondary structure change?

Line 132, The table number should be 1.

Line 136, Will be Coomassie brilliant blue-250.

Line 148, the characters for alpha-helix and beta-sheet should be supplied.

Line 155, RT should be specified in °C.

Author Response

Author's Reply to the Review Report (Reviewer 1)

This paper is dealing with the preparation and stability characterization of synthetic antigens to diagnose anti-chagas disease antibodies by ELISA. Although its importance is understandable, this topic is not related to the development of biosensor, at least in this stage. In general, immunoassays that require separation before measurement to obtain signal is not considered biosensor. Hence, here I just note some technical points to be considered in more suitable journal.

Question 1. Line 90 and 95, The authors state that the epitopes of IBMP antigen are linearized or linear, while its meaning is not clear. Does it mean that these proteins are recognized even after denaturation? If it is, why the authors use CD spectra to track its secondary structure change?

 Reply: There are indications that the epitopes are linear, as opposite to conformational 3D epitopes. However, all chimeric antigens have a strong CD signal. This signal is not typical of random coil, which would be the case if the chimeric proteins did not show any structural organization. Instead, the CD signal indicates that the chimeric antigens contain structural organization with some extend of secondary structures. Monitoring CD signal is a standard procedure to evaluate protein stability. Therefore, in this paper this method was used to evaluate the stability after heat- and time-induced denaturation. In order to avoid any misinterpretation, we have removed the word “linearized” from the manuscript, as follow:

Before: “Considering the predicaments herein set forth, our group expressed four linearized T. cruzi chimeric proteins...”

After: Considering the predicaments herein set forth, our group expressed four linearized T. cruzi chimeric proteins.

Before: “…that IBMP antigens are composed of linearized epitopes [10].”

After: …that IBMP antigens are composed of linear epitopes [10].

 Before: “Linearized antigens were purified affinity and ion-exchange…”

After: Linearized The antigens were purified affinity and ion-exchange…

Before: “Constitution of the IBMP linearized chimeric recombinant proteins.”

After: Constitution of the IBMP linearized chimeric recombinant proteins.

Question 2. Line 132, The table number should be 1.

Reply: We thank the reviewer for calling this to our attention. The sentence has been modified as follows:

Before: “Table 2. Constitution of the IBMP linearized chimeric recombinant proteins.”

After: Table 1. Constitution of the IBMP linearized chimeric recombinant proteins.

Question 3. Line 136, Will be Coomassie brilliant blue-250.

Reply: We thank the reviewer for calling this to our attention. The sentence has been modified as follows:

Before: “…and stained with Coomassie blue-250 for visualization.”

After: …and stained with Coomassie brilliant blue-250 for visualization.

Question 4. Line 148, the characters for alpha-helix and beta-sheet should be supplied.

Reply: We thank the reviewer for calling this to our attention. The sentence has been modified as follows:

Before: “…were used to assess - and -structural content, respectively.”

After: … were used to assess α- and β-structural content, respectively.

Question 5. Line 155, RT should be specified in °C.

Reply: We appreciate the reviewer’s concern regarding the clarity of room temperature and have specified RT as follows:

Before: “…and measurements were performed at RT.”

After: …and measurements were performed at RT (22-25 °C).

Reviewer 2 Report

The efficiency of serodiagnostic assays depends significantly from properties and stability of antigenic preparations. Antigens used for Chagas disease diagnosis have their specific properties and so need focused studies. The manuscript of Celedon et al. presents results of such study for some specially designed chimeric proteins, IBMP-8.1…8.4. In my opinion, the manuscript presents new and important results, but has very limited interceptions with biosensors. The main part of the study is focused on structural properties of the IBMP antigens. Their immunoreactivity was tested by the only technique, ELISA, which was implemented by routine protocol. The IBMP antigens can be used in different immunosensors, but these possible future activities are outside the article. Such MDPI journals as International Journal of Molecular Sciences or Molecules could be recommended for readdressing the manuscript.

Concerning the quality of the reviewed manuscript. It clearly presents the obtained results and needs only in few minor revisions.

  1. Overall data about Chagas disease (lines 43-49) should be accomplished by appropriate references.

  1. The concept of chimeric antigens has prolonged history and successful applications to different pathogens. However, the authors consider it in the Introduction (see lines 86-98) as a specific solution for Chagas disease antigens. The consideration of other efforts in this field (see DOI: 10.1038/s41467-018-03146-7; DOI: 10.1007/s10096-010-1010-3 as examples) will demonstrate extended perspectives for the use of the found regularities.

  1. In the course of ELISA testing the authors immobilized all antigens after their dilution in carbonate buffer (0.05 M, pH 9.6) – see line 166. This regime is really the most widespread in ELISA. However, considering different recombinant constructs and their possible denaturation, the authors should take in the mind that antigenic determinants could be better exposed in other solutions. This possibility should be at least indicated in the discussion.

  1. Many stabilizing compounds were proposed to decrease denaturation of stored proteins. However, the presented denaturation studies were held without such protectors. This decision could be reasonable taking into consideration further practical use of IBMP preparations, but actually it was not grounded in the manuscript.

  1. The Conclusions are limited by the statement of the simple fact that the IBMP-8.1…8.4 antigens diluted in carbonate buffer are very stable. This is a specific knowledge about the given preparations that could not be applied for other studies. Some generalized recommendations to simplify the work with other chimeric antigens will be useful here.

Author Response

Author's Reply to the Review Report (Reviewer 2)

The efficiency of serodiagnostic assays depends significantly from properties and stability of antigenic preparations. Antigens used for Chagas disease diagnosis have their specific properties and so need focused studies. The manuscript of Celedon et al. presents results of such study for some specially designed chimeric proteins, IBMP-8.1…8.4. In my opinion, the manuscript presents new and important results, but has very limited interceptions with biosensors. The main part of the study is focused on structural properties of the IBMP antigens. Their immunoreactivity was tested by the only technique, ELISA, which was implemented by routine protocol. The IBMP antigens can be used in different immunosensors, but these possible future activities are outside the article. Such MDPI journals as International Journal of Molecular Sciences or Molecules could be recommended for readdressing the manuscript. Concerning the quality of the reviewed manuscript. It clearly presents the obtained results and needs only in few minor revisions.

Question 1. Overall data about Chagas disease (lines 43-49) should be accomplished by appropriate references.

Reply: We thank the reviewer for calling this to our attention. The reference number [2] (Prata, A. Clinical and epidemiological aspects of Chagas disease. Lancet. Infect. Dis. 2001, 1, 92–100, doi:10.1016/S1473-3099(01)00065-2) has been included as follows:

Before: “Two distinct phases occur during the natural course of CD progression. The initial acute phase is characterized as an unspecific oligosymptomatic febrile illness. Infected individuals present high parasitemia, which enables the parasitological diagnosis based on the direct visualization of the parasite in a thick blood smear. The acute phase lasts for 2-3 months after the initial infection, followed by the gradual resolution of the clinical manifestations (when present) and the start of the lifelong chronic phase, characterized by an intermittent or absent parasitemia as well as high levels of IgG anti-T. cruzi antibodies.”

After: Two distinct phases occur during the natural course of CD progression. The initial acute phase is characterized as an unspecific oligosymptomatic febrile illness. Infected individuals present high parasitemia, which enables the parasitological diagnosis based on the direct visualization of the parasite in a thick blood smear. The acute phase lasts for 2-3 months after the initial infection, followed by the gradual resolution of the clinical manifestations (when present) and the start of the lifelong chronic phase, characterized by an intermittent or absent parasitemia as well as high levels of IgG anti-T. cruzi antibodies [2].

Question 2. The concept of chimeric antigens has prolonged history and successful applications to different pathogens. However, the authors consider it in the Introduction (see lines 86-98) as a specific solution for Chagas disease antigens. The consideration of other efforts in this field (see DOI: 10.1038/s41467-018-03146-7; DOI: 10.1007/s10096-010-1010-3 as examples) will demonstrate extended perspectives for the use of the found regularities.

 Reply: We thank the reviewer for this observation. The manuscript has been reviewed and the following sentence was added to the Introduction section:

Line 81-82: “Chimeric proteins have also been employed in IVD [13–15] or as a vaccine [16,17] for other infectious diseases.”

References:

13. Montagnani, F.; Paolis, F.; Beghetto, E.; Gargano, N. Use of recombinant chimeric antigens for the serodiagnosis of Mycoplasma pneumoniae infection. Eur. J. Clin. Microbiol. Infect. Dis. 2010, 29, 1377–1386, doi:10.1007/s10096-010-1010-3;

14. Lu, Y.; Li, Z.; Teng, H.; Xu, H.; Qi, S.; He, J.; Gu, D.; Chen, Q.; Ma, H. Chimeric peptide constructs comprising linear B-cell epitopes: application to the serodiagnosis of infectious diseases. Sci. Rep. 2015, 5, 13364, doi:10.1038/srep13364;

15. Beghetto, E.; Spadoni, A.; Bruno, L.; Buffolano, W.; Gargano, N. Chimeric antigens of Toxoplasma gondii: Toward standardization of toxoplasmosis serodiagnosis using recombinant products. J. Clin. Microbiol. 2006, 44, 2133–2140, doi:10.1128/JCM.00237-06.

16 . Hollingshead, S.; Jongerius, I.; Exley, R.M.; Johnson, S.; Lea, S.M.; Tang, C.M. Structure-based design of chimeric antigens for multivalent protein vaccines. Nat. Commun. 2018, 9, 1051, doi:10.1038/s41467-018-03146-7.

17. Nuccitelli, A.; Cozzi, R.; Gourlay, L.J.; Donnarumma, D.; Necchi, F.; Norais, N.; Telford, J.L.; Rappuoli, R.; Bolognesi, M.; Maione, D.; et al. Structure-based approach to rationally design a chimeric protein for an effective vaccine against Group B Streptococcus infections. Proc. Natl. Acad. Sci. 2011, 108, 10278–10283, doi:10.1073/pnas.1106590108.

Question 3. In the course of ELISA testing the authors immobilized all antigens after their dilution in carbonate buffer (0.05 M, pH 9.6) – see line 166. This regime is really the most widespread in ELISA. However, considering different recombinant constructs and their possible denaturation, the authors should take in the mind that antigenic determinants could be better exposed in other solutions. This possibility should be at least indicated in the discussion.

Reply: We thank the reviewer for his/her criticism, and we respectfully request that he/she reconsider this opinion. In a previous study performed by your group (DOI: 10.1371/journal.pone.0161100), we analyzed microplate sensibilization employing carbonate-bicarbonate (0.05 M, pH 9.6), sodium phosphate (0.05 M, pH 7.5) and MES ([2-(n-morpholino) ethanesulfonic acid] (0.05 M, pH 5.5). On that occasion, carbonate-bicarbonate (0.05 M, pH 9.6) was shown to be the most appropriate buffer system for reducing protein aggregation while maintaining the original conformation (under circular dichroism spectroscopy and dynamic light scattering analysis). This information is available in lines 199-204 (Results section) and in lines 278-294 (Discussion section), as follow:

Results section (line 199-204): “Since changes in CD spectra and light scattering may reflect structural shifts of the molecules, we previously characterized the IBMP chimeric antigens by these methodol-ogies to determine their typical conformation under different buffer compositions [10]. On that occasion, 50 mM carbonate-bicarbonate buffer pH 9.6 was shown to be the most appropriate buffer system for reducing protein aggregation while maintaining the orig-inal conformation.”

Discussion section (278-294): “Previously, our group characterized the secondary structure content and solubility of these IBMP chimeric antigen in different buffering agents: 50 mM carbonate-bicarbonate, pH 9.6; 50 mM sodium phosphate, pH 7.5; and 50 mM MES ([2-(n-morpholino) ethanesulfonic acid], pH 5.5) [10]. Those studies have demonstrated that IBMP-8.1 and IBMP-8.3 proteins predominantly comprised random coil as verified by neutral CD values between 215 and 240 nm and negative values at about 200 nm. On the other hand, CD spectra of IBMP-8.2 and IBMP-8.4 proteins exhibited a negative minimum at ~203 nm and shoulder at ~220 nm. The intensity of the CD signal at ~220 nm indicates that, in addition to random coil, IBMP-8.2 and IBMP-8.4 present also a certain content of α-helices. No conformational changes were observed in IBMP-8.1 and IBMP-8.3 coil proteins upon solubilization in different buffering agents. However, CD spectra of IBMP-8.3 and IBMP-8.4 proteins showed a slight shift to longer wavelengths upon solubilization in acidic pH, indicating changes in secondary structure content. With respect to DLS data, all proteins presented less polydisperse values (< 20%) in 50 mM carbonate-bicarbonate (pH 9.6) when compared to other buffers tested. These data indicate that all proteins can be influenced by the environment. Therefore, 50 mM carbonate-bicarbonate (pH 9.6) buffer system was chosen to carry out the present study.”

Question 4. Many stabilizing compounds were proposed to decrease denaturation of stored proteins. However, the presented denaturation studies were held without such protectors. This decision could be reasonable taking into consideration further practical use of IBMP preparations, but actually it was not grounded in the manuscript.

Reply: We thank the reviewer for his/her observation. The chimeric proteins already showed a high stability in the mild buffer used in this work, which basically consisted of the empirically determined pH and a low NaCl molarity.  These conditions were enough to maintain the recognition of the chimeric epitopes for the long-lasting storage. We have considered that adding stabilizing compounds, such as those used to stabilize therapeutic antibodies, would add up to the costs, as would have little effect on improving stability. Therefore, one of the aims of the work was to demonstrate that the chimeric antigens are stable in simple and low-cost buffer formulations. We also were looking for the ideal condition that may favor subsequent immunoassays.

Question 5. The Conclusions are limited by the statement of the simple fact that the IBMP-8.1…8.4 antigens diluted in carbonate buffer are very stable. This is a specific knowledge about the given preparations that could not be applied for other studies. Some generalized recommendations to simplify the work with other chimeric antigens will be useful here.

Reply: Unfortunately, it is not possible to make straightforward correlations between buffer composition and protein stability. In general, the buffer composition for high stability needs to be determined individually for each purified protein. The stability depends on the strength of its internal interactions and on the feature of the protein surface facing the solvent/buffer. These parameters are unique for each protein, as may be not the same in different environments too. For example, it is impossible to predict the tolerance of any cellular system expressing extraneous proteins, so empirical observations can led to the most appropriate chemical condition to solubilize the target protein. When using prokaryotic cultures, an option would be to manage a chemically mild cell lysis to separate the highly soluble protein fraction from possible inclusion bodies. These bodies are a by-product of cellular metabolism, but they may also be formed due to overexpression of heterologous proteins. If the recombinant protein is predominantly on the insoluble fraction, its in vitro stability will probably need of chaotropic agents, detergents or reducing agents. These chemicals are sometimes a detrimental element to subsequent bioassay platforms. The value of our finding is that the chimeric antigens are stable in buffer with simple composition, which is compatible with their use in immunoassay, without additional handling for long term storage.

Reviewer 3 Report

In this manuscript, the authors systematically analyzed the stability, structural integrity, and function of four IBMP chimeric proteins undergoing (1) heating (2) short-term room temperature storage (3) long-term 4 °C storage. This work addressed the unmet needs to understand the stability of IBMP chimeric antigens, and provided important information for the further application of these antigens as in vitro diagnostic test to identify chronic Chagas disease. Experiment design is logical, and data are convincing. In all, I recommend to accept this manuscript with some revision.

Comments:

  1. It would be great to have more than one test to analyze the function of IBMP. For example, liquid microarray test could be used in addition to the ELISA.
  2. The abbreviations of Chagas disease and circular dichroism used in this manuscript are the same — “CD”. it is confusing sometimes, especially when searching for a specific one.
  3. Page 4 line 132, table 2 —> table 1
  4. Page 4 line 148, alpha- and beta-
  5. table s2, workbook “ELISA”, time of exposure (months) —> (hours)

Author Response

Author's Reply to the Review Report (Reviewer 3)

In this manuscript, the authors systematically analyzed the stability, structural integrity, and function of four IBMP chimeric proteins undergoing (1) heating (2) short-term room temperature storage (3) long-term 4 °C storage. This work addressed the unmet needs to understand the stability of IBMP chimeric antigens, and provided important information for the further application of these antigens as in vitro diagnostic test to identify chronic Chagas disease. Experiment design is logical, and data are convincing. In all, I recommend to accept this manuscript with some revision.

 Question 1. It would be great to have more than one test to analyze the function of IBMP. For example, liquid microarray test could be used in addition to the ELISA.

Reply: The function of IBMP chimeric proteins in identifying anti-T. cruzi antibodies had already assessed by other methodologies, such as liquid microarray (10.1371/journal.pone.0161100 and 10.1128/JCM.00851-17), lateral flow assay (10.1155/2020/1803515), dual path platform (unpublished data), surface plasmon resonance (10.1016/j.bios.2020.112573), Western blot (manuscript in preparation) and double-antigen sandwich ELISA (manuscript in preparation). However, in the present study, we used ELISA because it is the most used methodology worldwide for chronic Chagas disease diagnosis. Liquid microarray is an expensive and laborious technique that is not used for routine diagnosis in most laboratories.

Question 2. The abbreviations of Chagas disease and circular dichroism used in this manuscript are the same — “CD”. it is confusing sometimes, especially when searching for a specific one.

Reply: We thank the reviewer for his/her observation and have abandoned the use of CD as an acronym for Chagas disease; all CD was replaced for Chagas disease and the acronym CD was used to refer only to Circular Dichroism.

Before: “Chagas disease (CD) is a deadly neglected tropical infection and…”

After (line 34): Chagas disease (CD) is a deadly neglected tropical infection and…

Before: “Two distinct phases occur during the natural course of CD progression.”

After (line 43): Two distinct phases occur during the natural course of Chagas disease progression.

Before: “Among the available commercial IVD tests to identify chronic CD...”

After (line 64): Among the available commercial IVD tests to identify chronic Chagas disease...

Before: “…parallel to diagnose CD in humans [9].”

After (line 71): …parallel to diagnose Chagas disease in humans [9].

Question 3. Page 4 line 132, table 2 —> table 1

Reply: We thank the reviewer for calling this to our attention. The sentence has been modified as follows:

Before: “Table 2. Constitution of the IBMP linearized chimeric recombinant proteins.”

After: Table 1. Constitution of the IBMP linearized chimeric recombinant proteins.

Question 4. Page 4 line 148, alpha- and beta-

Reply: We thank the reviewer for calling this to our attention. The sentence has been modified as follows:

Before: “…were used to assess - and -structural content, respectively.”

After: … were used to assess α- and β-structural content, respectively.

Question 5. table s2, workbook “ELISA”, time of exposure (months) —> (hours)

Reply: We thank the reviewer for calling this to our attention. The sentence has been modified as requested.

Round 2

Reviewer 1 Report

The authors made appropriate revisions to my technical questions. However, I am sorry but again I do not believe that this paper is dealing with biosensors. As in the Scope,

These biological recognition elements should be retained in close spatial contact with transducers including those based on the following principles:

  • electrochemical
  • optical
  • piezoelectric
  • thermal
  • magnetic
  • micromechanical

The journal will include a variety of subjects, including:

  • DNA chips
  • lab-on-a-chip technology
  • microfluidic devices
  • nanobiosensors and nanotechnology used in biosensors
  • biosensor fabrication
  • biomaterials
  • biosensor interfaces and membrane technology
  • in vitro and in vivo applications
  • instrumentation, signal treatment and uncertainty estimation in biosensors
  • drug discovery

However, the topic of ELISA antigen preparation is far from these recognition elements and subjects. 

Author Response

Reviewer question

The authors made appropriate revisions to my technical questions. However, I am sorry but again I do not believe that this paper is dealing with biosensors. As in the Scope: “These biological recognition elements should be retained in close spatial contact with transducers including those based on the following principles: electrochemical, optical, piezoelectric, thermal, magnetic, and micromechanical”. The journal will include a variety of subjects, including: DNA chips, lab-on-a-chip technology, microfluidic devices, nanobiosensors and nanotechnology used in biosensors, biosensor fabrication, biomaterials, biosensor interfaces and membrane technology, in vitro and in vivo applications, instrumentation, signal treatment and uncertainty estimation in biosensors, and drug discovery. However, the topic of ELISA antigen preparation is far from these recognition elements and subjects.

Reply: We appreciate the reviewer’s concern regarding the Scope of the journal; however, we respectfully believe this evaluation warrant reconsideration. Protein structural conformation, protein-protein interaction (e.g.: monomeric preservation, dimerization, etc) and stability assessment are paramount for an efficient immunoassay, whether it is applied in an ELISA system or microfluidic devices. All IBMP T. cruzi antigens were successfully employed for the detection of anti-T. cruzi immunoglobulins through a plethora of analytical systems, i.e. enzyme-linked immunosorbent assay, liquid microarray, immunochromatography/lateral flow immunoassay/dual path platform, Western blot, double sandwich ELISA and surface plasmon resonance. Bearing in mind that the formation of secondary structures, dimerization and agglutination negatively impacts the performance of an immunoassay and that antigens are the main gear driving indirect serological methods, analysis regarding an antigen’s behavior at different environmental conditions, temperatures and buffers paves the way towards a robust antigen preparation for a reliable and biosensor. As such, we believe that the manuscript is within the subjects covered by the Biosensors journal and warrants consideration for publication.